# Efficient interatomic descriptors for accurate machine learning force fields of extended molecules

Adil Kabylda[1], Valentin Vassilev-Galindo [1], Stefan Chmiela [2,3], Igor Poltavsky[1] & Alexandre Tkatchenko [1]✉

Machine learning force fields (MLFFs) are gradually evolving towards enabling molecular dynamics simulations of molecules and materials with ab initio accuracy but at a small fraction of the computational cost. However, several challenges remain to be addressed to enable predictive MLFF simulations of realistic molecules, including: (1) developing efficient descriptors for non-local interatomic interactions, which are essential to capture long-range molecular fluctuations, and (2) reducing the dimensionality of the descriptors to enhance the applicability and interpretability of MLFFs. Here we propose an automatized approach to substantially reduce the number of interatomic descriptor features while preserving the accuracy and increasing the efficiency of MLFFs. To simultaneously address the two stated challenges, we illustrate our approach on the example of the global GDML MLFF. We found that non-local features (atoms separated by as far as 15 Å in studied systems) are crucial to retain the overall accuracy of the MLFF for peptides, DNA base pairs, fatty acids, and supramolecular complexes. Interestingly, the number of required non-local features in the reduced descriptors becomes comparable to the number of local interatomic features (those below 5 Å). These results pave the way to constructing global molecular MLFFs whose cost increases linearly, instead of quadratically, with system size.

Reliable atomistic force fields are essential for the study of dynamics, thermodynamics, and kinetics of (bio)chemical systems. Machine learning force fields (MLFFs) are lately becoming a method of choice for constructing atomistic representations of energies and forces[1–18]. Contrary to traditional computational chemistry methods, MLFFs use datasets of reference calculations to estimate functional forms which can recover intricate mappings between molecular configurations and their corresponding energies and/or forces. This strategy has allowed to construct MLFFs for a wide range of systems from small organic molecules to bulk condensed materials and interfaces with energy prediction errors below 1 kcal mol$^{-1}$ with respect to the reference ab initio calculations[1–3,18–27]. Applications of MLFFs already include understanding the origins of electronic and structural transitions in materials[20], computing molecular spectra[21–24], modeling chemical reactions[25], and modeling electronically excited states of molecules[26,27]. Despite these great successes of MLFFs, many open challenges remain[18,28,29]. For instance, the applicability of MLFF models to larger molecules is limited, partly due to the rapid growth in the dimensionality of the descriptor (i.e., a representation used to characterize atomic configurations).

A descriptor used to encode the molecular configurations determines the capability of an MLFF to capture the different types of

[1]Department of Physics and Materials Science, University of Luxembourg, L-1511 Luxembourg City, Luxembourg. [2]Machine Learning Group, Technische Universität Berlin, 10587 Berlin, Germany. [3]BIFOLD – Berlin Institute for the Foundations of Learning and Data, 10587 Berlin, Germany. ✉e-mail: alexandre.tkatchenko@uni.lu

interactions in a molecule. Therefore, descriptors are designed to contain features that emphasize particular aspects of a system or to highlight similar chemical or physical patterns across different molecules or materials. Many different descriptors have been proposed to construct successful MLFFs for specific subsets of the vast chemical space[2,6,30–39]. However, there is no guarantee that a given descriptor is capable of accurately describing all relevant features throughout high-dimensional potential-energy surfaces (PESs) that characterize flexible molecular systems[28]. The main challenge here is to balance the number of features required for a given ML model to describe simultaneously the interplay between short and long-range interactions. One possible approach to address this challenge is to increase the complexity of descriptors by adding explicit features to model-specific interactions[28,40]. However, such a solution usually yields descriptors that are high-dimensional and inefficient for large systems. As an alternative solution, several approaches have been proposed to generate reduced descriptors, targeting specific properties of interest[41–45]. Such reduced descriptors have led to insights into complex materials, and this approach has also been applied to MLFFs, specifically to reduce ACSFs and SOAP representations[46–49].

The descriptors discussed above correspond to local MLFF models, where only a certain neighborhood of atoms is considered within a specified cutoff distance. Such locality approximation is usually employed in MLFFs to enhance their transferability and applicability for larger systems than the given training set. However, as a downside, accounting for long-range interactions requires additional effort. Therefore, some recent MLFF models[12,21,40,50–53] have integrated correction terms to account for certain long-range effects (e.g., electrostatics), but long-range electron correlation effects are still not well characterized. It is evident that the field of MLFF combined with physical interaction models is rapidly growing and developing, but a definitive solution to these challenges has not yet been found. In general, ML models should be able to correctly describe (i) the non-additivity of long-range interactions, (ii) the strong dependence of such interactions on the environment of interacting objects, and (iii) the non-local feedback effects that lead to a multiscale nature of long-range interactions. Addressing these features requires developing flexible—and simultaneously accurate and efficient—MLFFs without employing strictly predefined functional forms for interactions or imposing characteristic length scales.

Alternatively, one can switch to so-called global descriptors, such as the Coulomb matrix, where all interatomic distances are considered. Unfortunately, such global descriptors scale quadratically with system size. In addition, reducing the descriptor dimensionality in global models is an unsolved challenge. For example, it is evident that most short-range features (e.g., covalent bonds, angles, and torsions) should be preserved when constructing accurate MLFFs. In fact, the number of local features scales linearly with the system size. In contrast, the number of non-local (long-range) features scales quadratically and a general coarse-graining procedure to systematically reduce non-local features does not exist yet.

To address these challenges, in this work, we propose an automatic procedure for identifying the essential features in global descriptors that are most relevant for the description of large and flexible molecules. We apply the developed approach to identify efficient representations for various systems of interest, including a small molecule, a supramolecular complex, and units of all four major classes of biomolecules (i.e., proteins, carbohydrates, nucleic acids, and lipids): aspirin (21 atoms), buckyball catcher (148 atoms), alanine tetrapeptide (Ac-Ala3-NHMe, 42 atoms), lactose disaccharide (45 atoms), adenine-thymine DNA base pairs (AT-AT, 60 atoms), and palmitic fatty acid (50 atoms). Employing the reduced descriptor results in an improvement in prediction accuracy and a two- to four-fold increase in computational efficiency. Moreover, an analysis of the features that are selected by our reduction procedure suggests that these features follow certain patterns that are explained by both interaction strength and statistical information they provide about atomic fluctuations. In particular, while most short-ranged features are essential for the PES reconstruction, a linearly scaling number of selected non-local features are enough for an ML model to describe collective long-range interactions.

## Results

The quadratic scaling of global descriptors with molecular size, especially their long-range part, becomes a considerable challenge with the increasing number of atoms. For molecules containing just a few dozen of atoms, such descriptors are, in fact, substantially over-defined. For example, the number of degrees of freedom (DOF) uniquely defining a configuration of a molecule with $N$ atoms is $3N - 6$. At the same time, the Coulomb matrix and related global descriptors contain $N(N-1)/2$ DOFs. Thus, such descriptors will span a much larger space than what is effectively needed, making ML models harder to optimize and compromise their performance/accuracy. In the case of a complete interatomic inverse distances descriptor (a simplified version of the Coulomb matrix[31]), the interatomic interactions can be visualized as a fully connected graph with atoms as nodes and descriptor features as edges. For example, Fig. 1a shows such a descriptor for the Ac-Ala3-NHMe molecule containing 861 features. Each edge of the graph represents a dimension in the descriptor space, where an ML model should be trained.

The large dimensionality of the descriptor significantly complicates the learning task. The interaction map (Fig. 1b) shows how the (s) GDML model interprets the interatomic interactions when the entire global interatomic inverse distance descriptor is employed (values are averaged over 1000 configurations, see Methods for further details). As a projection of complex many-body forces into atomic components, this partitioning is non-unique and is mainly determined by the chosen descriptor. In turn, the simpler the descriptor space, the more straightforward the task for the ML model. One can see that the interaction map shown in Fig. 1b is rather non-uniform and complex, meaning the (s)GDML model needs to be able to reproduce a complex mapping between the descriptor (861 dimensions) and force (126 dimensions) spaces.

### Reduced descriptors

The automatized descriptor reduction procedure proposed in this work significantly simplifies the learning task and noticeably decreases the complexity of the interaction map. To reduce the size of the descriptor, we employ a definition of similarity between system states, which plays a pivotal role in kernel-based ML models. Namely, we assume that the least important descriptor features for the similarity measure can be omitted without losing generality in an MLFF model (see Methods for further details). The reduced descriptor space of the optimal ML model (344 features) is shown as a graph in Fig. 1c. Interestingly, the short-range part (236 features) of the graph is practically unaltered by the reduction procedure. In contrast, a small fraction of long-range features (108 out of 574 in the full descriptor) enables an accurate account of all relevant long-range forces while greatly simplifying the interaction map (Fig. 1d). The reduced descriptor still completely and uniquely represents the molecular configurations. For Ac-Ala3-NHMe, we can remove up to 60% of the initial global descriptor while preserving the accuracy of the (s)GDML model (Fig. 1e). This is a remarkable result since many approaches for reducing the dimensionality of the learning task (e.g., low-rank approximations of the kernel matrix) typically lead to performance degradation because the model has to compensate for omitted features in some arbitrarily reduced representation[54].

We also analyzed the features that are kept to interpret the content of the reduced descriptor (Fig. 1f). One sees that the reduced descriptors are not a simple localization because features' importance

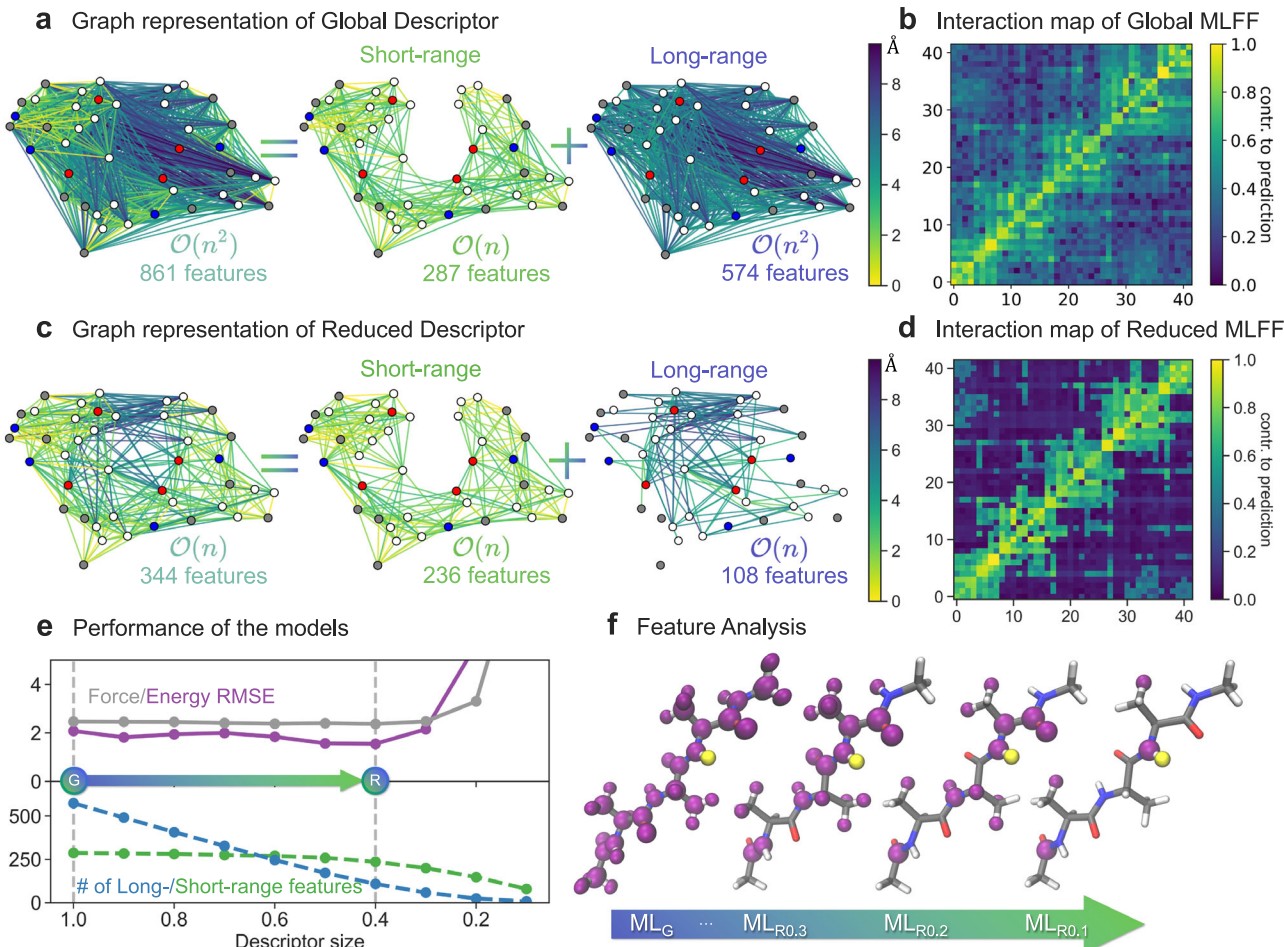

**Fig. 1 | Overview of the descriptor reduction scheme. a, c** Graph representation of global and reduced descriptors for Ac-Ala3-NHMe, and its decomposition into short- and long-range features with corresponding scaling with the number of atoms, $n$. The color of nodes indicates atom type: H - white, C - gray, N - blue, O - red. The color of the edges indicates the average distance between atoms. **b, d** Interaction map of global and reduced Machine Learning Force Fields (MLFFs). Each square in the heatmaps represents a given pair of atoms in the molecule (atom indices start from 0). The colorbar is in log scale normalized to the 0-1 range and goes from dark magenta (small) to yellow (large) for the average contributions to the force prediction. **e** Performance of the global and reduced models: energy (in kcal mol⁻¹) and force (in kcal (mol Å)⁻¹) root mean square errors (RMSEs) as a

function of the size of the descriptor (upper panel). The RMSE values were calculated on a test set of ~80k points, distinct from the training (1k) and validation sets (1k). Decomposition of the descriptor into short- and long-range features (lower panel). The arrow between two vertical dashed lines highlights the degree of reduction between global (G) and optimally reduced (R) models. Descriptor sizes in $x$ axis go from 1 to 0, where 1 corresponds to a default global descriptor and 0 to an empty descriptor. **f** Feature analysis in the global (denoted as ML$_G$) and reduced models (denoted as ML$_{RX}$, where X indicates the descriptor size). Hydrogen atom highlighted in yellow keeps interactions with atoms highlighted in purple. The arrow indicates transition of the employed descriptor from default global to substantially reduced ones.

is not necessarily correlated with the distance between atoms. The proposed selection scheme considers both the strength of the interactions between atoms and the information the features provide about the molecular structure. The latter means that the optimal non-local features depend on the training dataset and the respective sampled region of PES. As a possible future outlook, one could consider switching from selecting atom-centered non-local features from the initial global descriptor to projecting them into more efficient and general collective coordinates. This would provide us with effective interaction centers for large molecules, similar to those employed by the TIP4P[55] or Wannier centroid[53] models of water. In turn, this would enable the construction of automatized coarse-grained representations preserving the MLFFs accuracy, a long-desired tool for simulating complex and large systems.

The proposed descriptor reduction scheme is general and applicable to a wide range of systems. Figure 2 shows GDML performance curves of energy and forces for aspirin, Ac-Ala3-NHMe, AT-AT, and the buckyball catcher as a function of the size of the descriptor for different sizes of the training set. The aspirin molecule represents a

rather small semi-rigid molecule, for which one can already build accurate and data-efficient MLFFs[1-3,12,13,34,56]. The other molecules represent large and flexible systems that constitute a challenge for existing ML models. For each of these systems, GDML models with 300, 500, 800, and 1000 training points were trained using descriptors of different sizes. For the Ac-Ala3-NHMe molecule, due to its size and flexibility, we have also constructed the model using 3000 training points.

For a small molecule such as aspirin (210 features in the original descriptor), the descriptor showing the lowest RMSEs is the default global descriptor. Nevertheless, removing up to 30% of the descriptor only slightly affects the predictions of the model. Whereas, for Ac-Ala3-NHMe (861 features), AT-AT (1770 features), and the buckyball catcher (10878 features) one can significantly reduce the size of the descriptor while obtaining even more reliable predictions regardless of the training set size (Fig. 2b–d). For instance, models trained on 1000 training samples with a descriptor size reduced by 60% provide energy and force RMSEs that are up to 2.2 kcal mol⁻¹ and 0.2 kcal (mol Å)⁻¹ lower than those of the models employing default global descriptors.

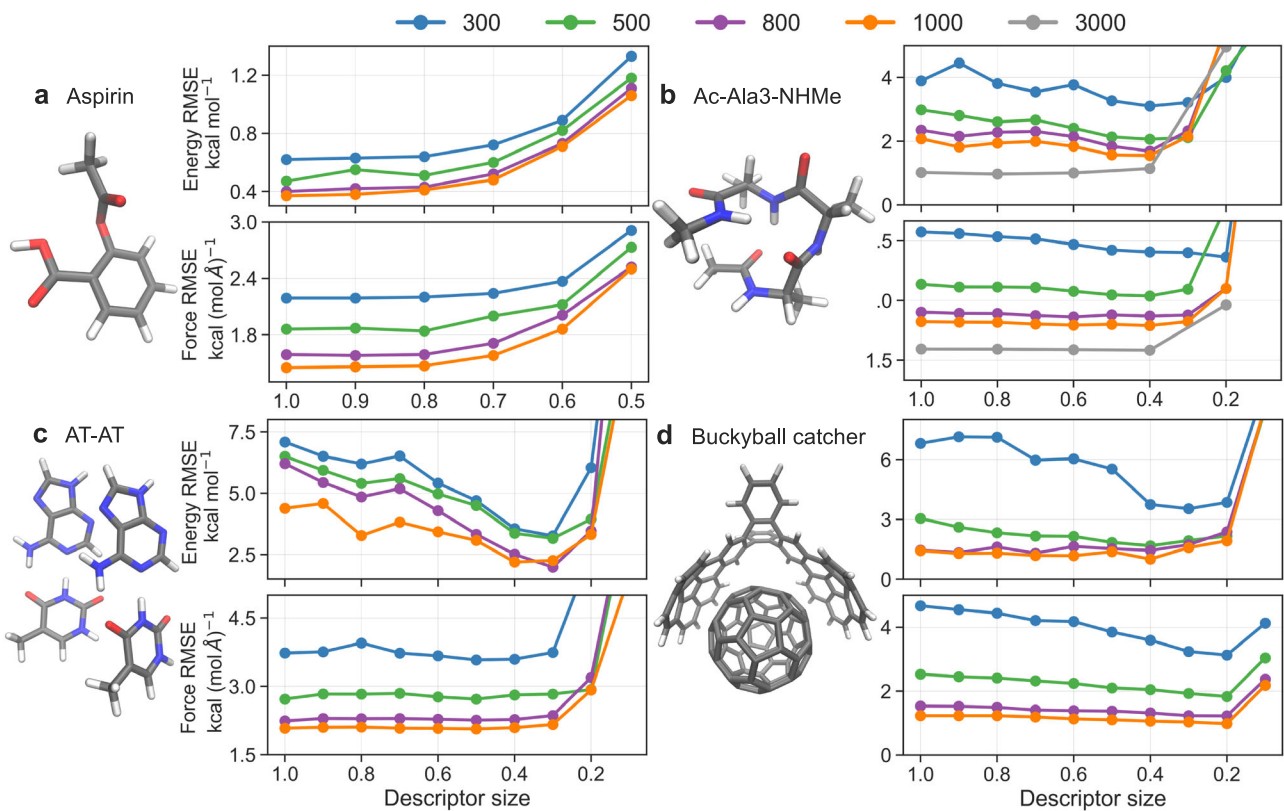

**Fig. 2 | Accuracy of the models with reduced descriptors.** Energy (in kcal mol⁻¹) and force (in kcal (mol Å)⁻¹) root mean square errors (RMSEs) as a function of the size of the descriptor. RMSEs of Gradient Domain Machine Learning (GDML) models for aspirin (**a**), Ac-Ala3-NHMe (**b**), AT-AT (**c**), and the buckyball catcher (**d**) trained on 300, 500, 800, 1000, and 3000 configurations. Descriptor sizes in $x$ axis go from 1 to 0, where 1 corresponds to a default global descriptor and 0 to an empty descriptor.

The different behavior in prediction accuracy with decreasing size of the descriptor between aspirin and other bigger molecules is mainly caused by the differences in their size. Indeed, with increasing molecule size, the quadratic redundancy of the feature space offers greater reduction potential. Therefore, reducing the number of features contained in a global descriptor should be a routine task for building ML models of large molecules.

**Improved description of interactions**

The improved accuracy of models trained using reduced descriptors is a consequence of how well those models describe the interatomic interactions. Figure 3 shows the interaction heatmaps and interatomic-distance heatmaps averaged over 1000 conformations for Ac-Ala3-NHMe, AT-AT, and the buckyball catcher. For each of the molecules, we use the following GDML models trained with 1000 configurations: (i) the ML$_{Global}$ model, (ii) a model trained using a $\frac{1}{r}$ descriptor mimicking a local descriptor by removing all features involving distances greater than 5.0 Å (the typical value for the cutoff radius in local descriptors) in at least one configuration in the dataset (ML$_{Local}$), and iii) a ML$_{Reduced}$ model. We remark that the prediction accuracy of our reduced models for large molecules is superior to state-of-the-art kernel-based local GAP/SOAP[8] ML model (see Supplementary Note 1 and Supplementary Fig. 1).

For the ML$_{Global}$ models (containing 861 features for Ac-Ala3-NHMe, 1770 for the AT-AT, and 10878 for the buckyball catcher) the contributions are evenly distributed among different pairs of atoms regardless of the distance between them. This allows the model to effectively capture long-range interactions, but as a downside may degrade the ability to optimally resolve all short-range ones. Conversely, the ML$_{Local}$ models (with a size equal to

33%, 17%, and 15% of the size of the default global descriptor for Ac-Ala3-NHMe, AT-AT, and the buckyball catcher, respectively) only rely on the local environment of the molecule. This is confirmed by the contributions of the atoms to the force prediction of other atoms, which are directly related to the magnitude of the corresponding interatomic distances. Thus, the ML$_{Local}$ models offer a more adequate description of short-range interactions but completely neglect those interactions arising from distances greater than the selected cutoff. One of the drastic consequences of such neglect is the instability of MD simulations performed using these local MLFFs. Finally, the ML$_{Reduced}$ models offer an improvement over both ML$_{Global}$ and ML$_{Local}$ models by achieving an adequate description of the local environment of the molecule and, at the same time, keeping the relevant information for describing non-local interactions. Therefore, using a reduced descriptor leads to ML models that provide a balanced, faithful description of all essential interactions in a given system.

We further compare the transferability of the global and reduced models by training them on compact structures and testing on extended structures of the tetrapeptide (and vice-versa). To do that, we split the tetrapeptide dataset based on the distance between the furthest atoms (ranges from ~8 to ~14 Å) with a threshold of 12 Å (and 9.5 Å, see Supplementary Note 2). The comparison of the force and energy RMSEs shows that the reduced models are more accurate than global models when dealing with unseen outlier extended or compact structures of the tetrapeptide (see Supplementary Table 1). This suggests that an overdetermined global ML model underperforms due to conflicting information from excessive features and that the model with a reduced descriptor indeed provides an improved description of interactions.

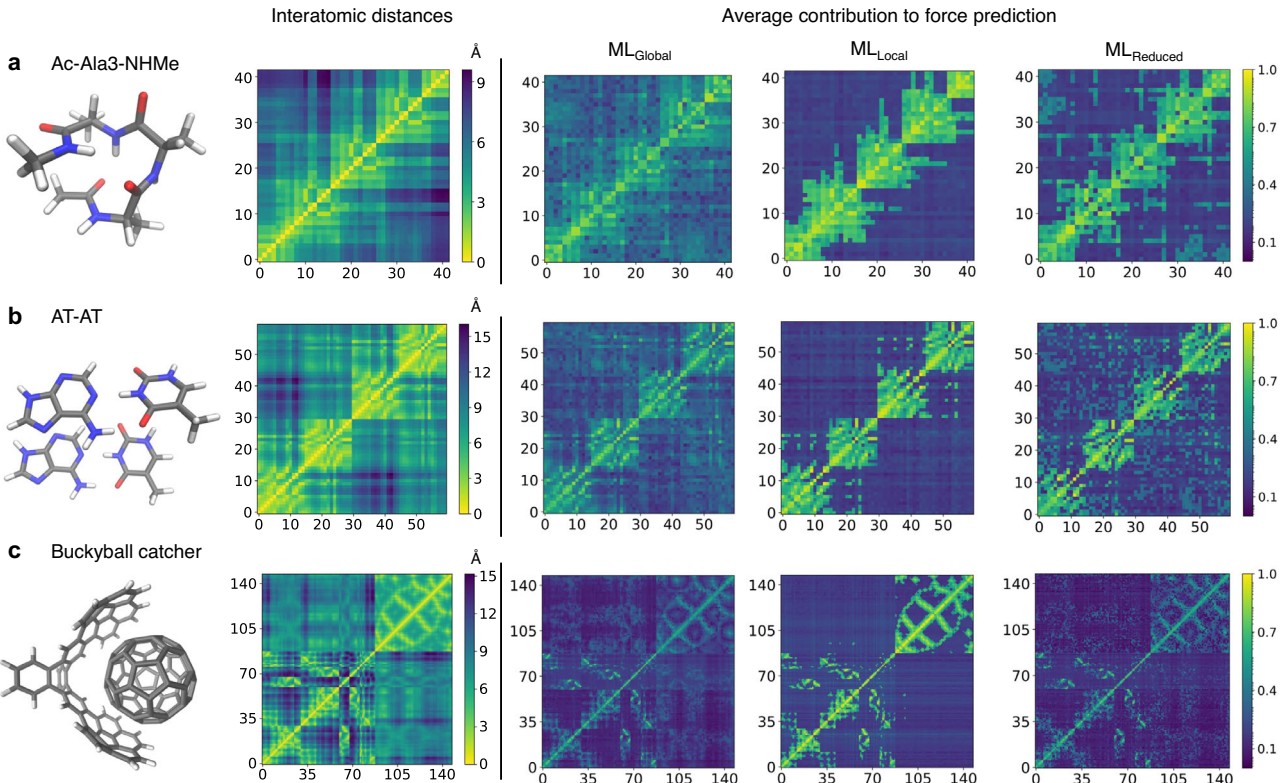

**Fig. 3 | Complexity of interaction patterns.** Heatmaps of average interatomic distances (in Å) and average contributions (in log scale normalized to the 0-1 range) of each atom to the force prediction of all atoms computed from 3000 configurations of Ac-Ala3-NHMe (**a**), AT-AT (**b**), and 1000 configurations of the buckyball catcher (**c**). Each square in the heatmaps represents a given pair of atoms in the molecule (atom indices start from 0). The scale goes from yellow (short distances) to dark magenta (long distances) for interatomic distances, and from dark magenta (small contributions) to yellow (large contributions) for the contributions to the force prediction.

## Efficiency of reduced-descriptor models and stability of molecular dynamics

The models obtained using reduced descriptors, together with the increment in accuracy provide up to a ten-fold increase in efficiency during training and four-fold during deployment (Supplementary Table 2). Improvement in efficiency results from the fact that there are less noisy features in the reduced model, which leads to lower per-iteration costs. For training, such efficiency can only be obtained with a recently developed iterative solver[57], while the evaluation speedup is always present when using the GDML model.

We also checked the stability of molecular dynamics simulations employing reduced models. We found that optimally reduced models for Ac-Ala3-NHMe ($ML_{R0.6}$ with 3000 training points, 0.3 fs timestep) and the buckyball catcher ($ML_{R0.2}$ with 1000 training points, 0.5 fs timestep) are stable and the corresponding energy is conserved during the dynamics at 300 K for 3 ns.

In complex systems, long simulations can be unstable due to incomplete dataset even with the default global descriptor. For example, AT-AT show degraded stability when encountering rare/new configurations that are not well sampled in the dataset (decomposes due to leaving the planar configuration or due to hydrogen transfer from T to A). Still, we find that simulations can remain stable for 3 ns with the default $ML_{R1.0}$ and the reduced $ML_{R0.5}$ models (1000 training points, 0.1 fs timestep). Further stability depends on the accuracy of the underlying original model. Thus, with increasing complexity of the PES, one should consider using active learning to detect "dark" states and adding them to the training process regardless of the employed descriptor.

Models with substantially reduced descriptors can only describe a smaller part of the PES and lead to artificial behavior

(e.g., steric clashes or fragmentation). Such artifacts happen with a higher probability in flexible molecules where atom pairs corresponding to removed features might come close, and their relative position cannot be neglected anymore. For example, we encounter steric clashes in Ac-Ala3-NHMe when using $ML_{R0.4}$ trained on 1000 configurations at 0.5 fs timestep, even though test errors are lower than those of the global $ML_{R1.0}$ model. Therefore, smaller prediction errors do not always lead to a more reliable ML model when tested in an extended simulation of several nanoseconds (see also ref. 58).

In order to further demonstrate the stability and the broad applicability of reduced GDML models, we study the evolution of the tetrapeptide molecule from a compact to an extended structure under a constant external force of 10 pN applied in opposite directions to the two terminal carbon atoms (Fig. 4). Statistics were collected using 30 simulations with different initial velocities following the Boltzmann distribution at 300K (Supplementary Fig. 3). We ran simulations using the global and reduced models trained with 5000 training points. In addition, we ran simulations at two levels of theory - PBE and PBE+MBD - and used the resulting data for validation (see Methods for further details). We measured the structural compactness using the gyration radius and compared the dynamical properties of the models. As expected, due to the absence of attractive dispersion interactions in the PBE simulations, the tetrapeptide unfolded faster than in the PBE+MBD ones (on average it took ~550 and ~750 fs, respectively, to reach $R_g$ = 3.8 Å). Both the global and reduced models agreed well with the PBE+MBD results, indicating their accuracy and reliability. Also, this confidently shows that the reduced model preserves all the information needed to describe long-range interactions with ab initio accuracy.

After confirming the reliability of the reduced model, we further investigated the conformational space of the tetrapeptide to enhance our understanding of its behavior. To achieve this, we conducted multiple simulations in parallel, with an accumulated time of 50 ns (Supplementary Fig. 4). This approach allowed us to obtain a converged folding and unfolding distribution, which was visualized using the $\psi_2$ angle in the probability distributions for the central residue. Our

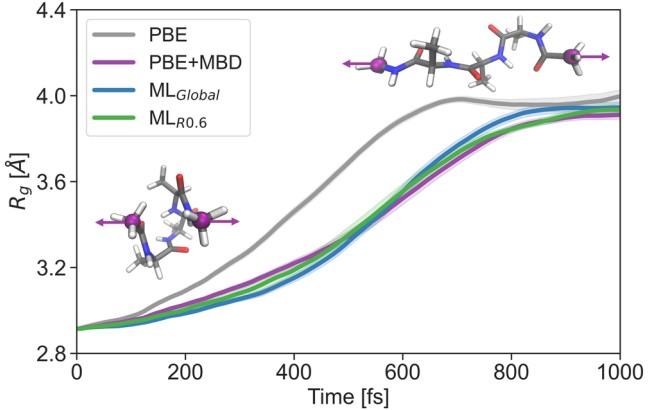

**Fig. 4 | Steered dynamics between folded and extended states of the tetrapeptide.** The tetrapeptide undergoes unfolding due to an external force acting parallel to the connecting line between two terminal carbon atoms. The gyration radius, averaged over 30 runs, is represented by the solid lines, with the shaded areas indicating the standard error.

analysis reveals that the tetrapeptide populates the extended state with a probability of 13% on the 50 ns time scale (Supplementary Fig. 5).

## Relevance of interatomic descriptor features

The importance of descriptor features is not always related to the magnitude of their contribution to the model predictions (Fig. 5a). As expected, the features contributing the most to the force predictions (above 0.5–0.6 a.u.) in the global model are all included in the reduced descriptor (see the top marginal plots). These strongly contributing features are primarily associated with short interatomic distances. In contrast, the selected features corresponding to medium- and long-range interatomic distances span almost all contribution ranges. For instance, some weakly contributing features that describe an average distance as large as 15 Å are included in the reduced descriptor of the AT-AT system. The distribution of the selected features is skewed towards the weak contributions upon increasing the molecular size (compare gray density distribution of three molecules in top marginal plots, Fig. 5a). Interestingly, the distribution of the contribution of the selected features is significantly shifted towards larger values after retraining the ML model (center marginal plots, Fig. 5a).

Further analysis reveals that contribution of particular features in the global model can range from linear to stochastic with respect to the interatomic distance (Supplementary Fig. 6a). The proportion of stochastic features increases with the size of the systems and the size of training set (Supplementary Fig. 6b). In the reduced models after retraining most of the selected features have a high coefficient of determination, $R^2$ (Supplementary Fig. 6c). Contribution of "linear" features to the force prediction decrease quadratically with distance

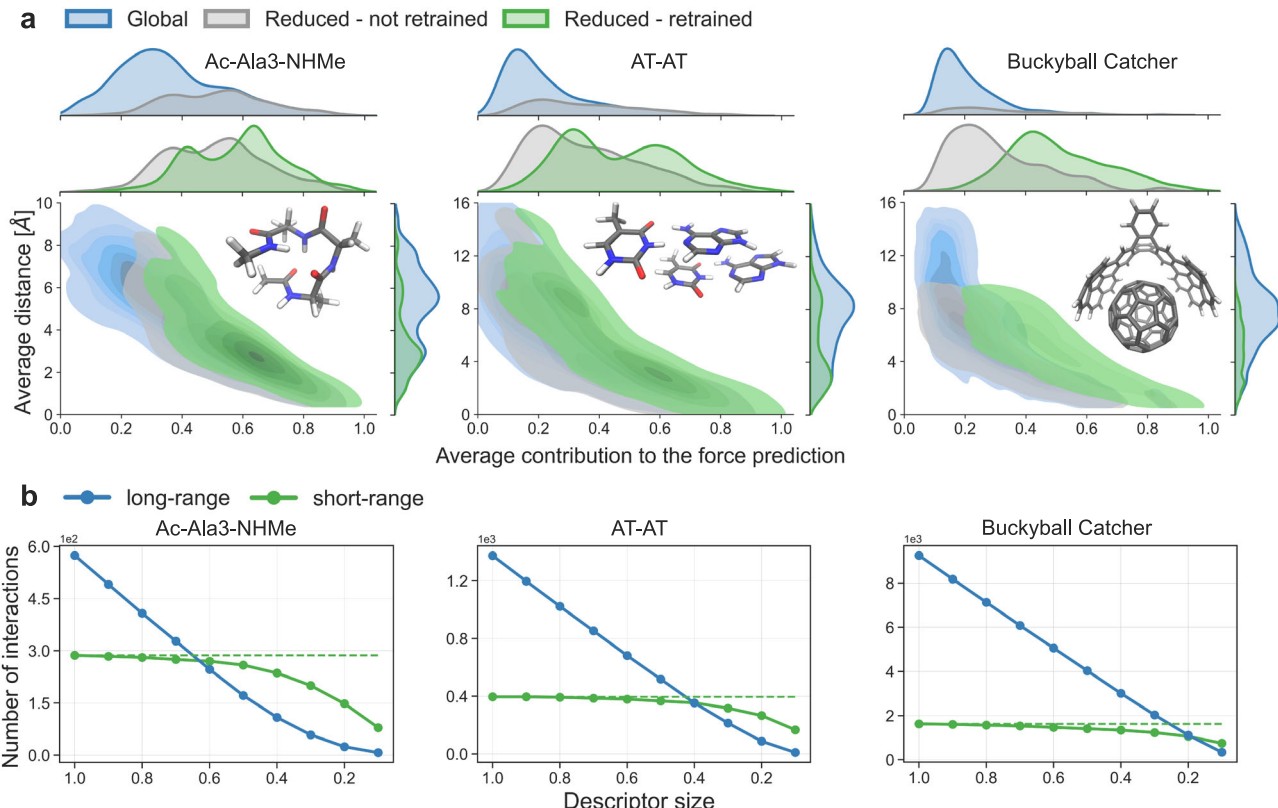

**Fig. 5 | Analysis of relevant interatomic features. a** Distributions of the average distance of pairwise features and average contribution of features to the force prediction for Ac-Ala3-NHMe, AT-AT, and the buckyball catcher using bivariate kernel density estimate plots (Machine Learning models: global - green, reduced before retraining - gray, reduced after retraining - green). The marginal charts on the top and right show the distribution of the two variables using density plot. The average values were obtained from all configurations in the datasets. The *x* axis is in

log scale normalized to the 0–1 range. **b** Decomposition of the reduced descriptor by short- and long-range features for Ac-Ala3-NHMe, the AT-AT, and the buckyball catcher. Pairwise features with an average distance below 5 Å across all configurations in the dataset are counted as short-range (green line), long-range otherwise (blue line). Dashed green line represent number of short-range features in the global descriptor. Descriptor sizes in *x* axis go from 1 to 0, where 1 corresponds to a default global descriptor and 0 to an empty descriptor.

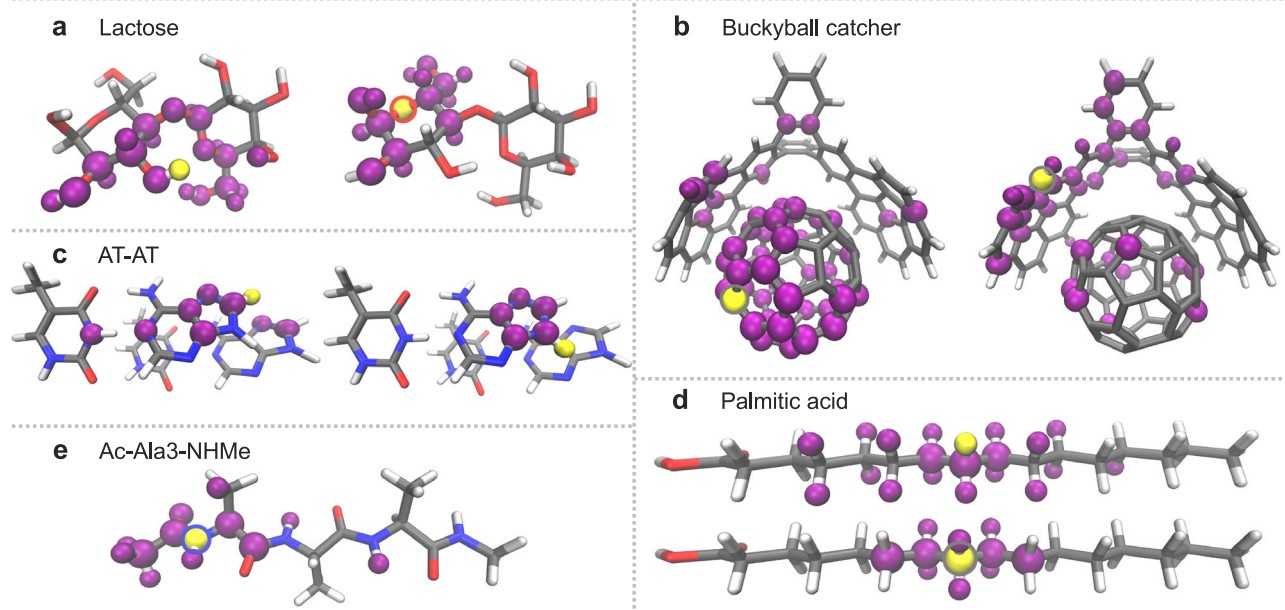

**Fig. 6 | Examples of features in the reduced descriptors.** The features were obtained from the reduced ML$_{R0.3}$ descriptor for lactose (**a**), AT-AT (**c**), palmitic acid (**d**), and Ac-Ala3-NHMe (**e**); the ML$_{R0.2}$ descriptor for buckyball catcher (**b**). Root mean square errors as a function of the descriptor size for lactose and palmitic acid can be found in Supplementary Fig. 2. Atoms highlighted in yellow keep in the reduced descriptor the features that correspond to interactions with purple atoms. Outline colors on reference atoms highlighted in yellow indicate their chemical symbols (hydrogen - no outline, carbon - gray, nitrogen - blue, oxygen - red).

(slope ≈ −2), suggesting the prominence of Coulombic contributions to the interatomic forces.

These findings are general and valid for all descriptor reduction degrees. Although we do not rely on any characteristic lengthscale, we show the effect of descriptor reduction approach on different types of interactions as conventionally defined when imposing length scales. Figure 5b shows a decomposition of the reduced descriptor in short- and long-range features for different descriptor sizes. We consider the feature as short-ranged if the distance between two atoms across all configurations in the dataset is below 5 Å, and long-ranged otherwise. In all the cases, feature selection removes prevalently long-range features and 10–20% of the local features that always lie under 5 Å (compare dashed and solid green lines in Fig. 5b). Nevertheless, we emphasize that removing local features might worsen the stability in flexible systems and the best-practice solution is to keep them all in the reduced descriptor.

We construct our datasets using dynamics simulations at the PBE +MBD level of theory (see Methods for further details). However, long-range descriptor features are also kept in PBE calculations, as the PBE functional includes both long-range electrostatics and polarization, despite the semi-local nature of the exchange-correlation term. When we account for MBD, up to 22% of removed features can change depending on the degree of reduction (~5% for the optimal ML$_{R0.4-0.6}$ models). This is consistent with the fact that MBD contribution to the energy is relatively small compared to the PBE energy. Nevertheless, MBD contribution can greatly influence the dynamics of chemical systems, particularly in the case of large and flexible molecules. For example, MBD is essential for accurately evaluating the stability of aspirin polymorphs[59], standing molecules on surfaces[60], and interlayer sliding of 2D materials[61]. Therefore, it is essential to perform PBE+MBD calculations in order to generate a reliable dataset in the first place.

**Analysis of patterns in relevant interatomic features**
We analyze particular atoms and their selected chemical environment in the reduced descriptors to identify the trends in the features that an ML model considers essential. To make our results general for a wide range of (bio)molecules, we include lactose and palmitic acid in our discussion. Figure 6 shows examples of such features for all of our test systems. One can see some general patterns. For example, shielded atoms (C, N, O) in backbone chains usually keep solely local features (Fig. 6a, e, d). Most interactions between the first-, second-, and third-nearest neighbors are intact. Such behavior is expected and reflects the importance of the local environment in describing interatomic interactions.

The outer atoms are responsible for accounting for the relevant non-local features in the molecules (Fig. 6c, d, and Fig. 1f). The flexibility of the molecule defines the number of such features in the reduced descriptor. For instance, outer hydrogen atoms in semi-rigid molecules (e.g., lactose) only require local information in the descriptor. In contrast, flexible molecules (e.g., Ac-Ala3-NHMe and palmitic acid) present a combination of short-range features to describe local bond fluctuations and a substantial number of non-local features for accurately characterizing essential conformational changes, such as the folding and unfolding of peptide chains (Figs. 1f and 6d).

There are more complex patterns like those observed in the AT-AT base pairs and the buckyball catcher (Fig. 6c, b). In the former, two hydrogen atoms in the imidazole ring of adenine retain contrasting sets of features. This is because some features contained in an optimal descriptor depend on the phenomena sampled in the datasets (e.g., MD trajectories at certain temperatures). In the buckyball catcher, the reduced descriptor reveals that the symmetry of the system is important. Only a few features from the catcher are needed to effectively describe the interaction with any atom in the buckyball (and vice-versa).

**Linear scaling of descriptors with molecular size**
As a result of the descriptor reduction procedure, we obtain reduced descriptors that scale linearly with the number of atoms (Fig. 7). This is achieved by revealing a minimal complete set of non-local features that describe long-range interactions. The number of such features is similar to the number of short-range ones (Fig. 5b). Therefore, reduced descriptors not only scale linearly with the system size but also the corresponding prefactor (~10) is a few orders of magnitude smaller to

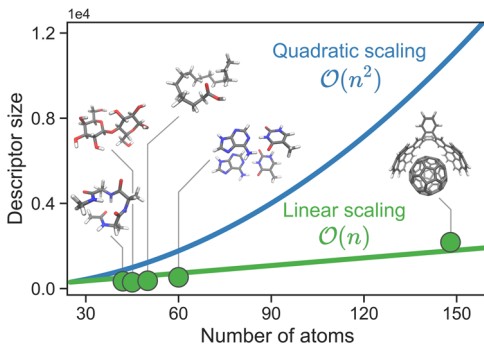

**Fig. 7 | Scaling of the default and reduced global descriptors.** Dots represent reduced descriptors for the molecules used in this study.

that of local descriptors (~1000). However, we must note that the Hessian of the GDML model is still of the same size ($3N \times 3N$, where $N$ is the number of atoms), though many of the entries are omitted in the reduced model (as shown in Fig. 3). This noticeably reduces the computational cost of global ML models (up to a factor of four for studied systems) and paves the road to constructing efficient global MLFFs for systems consisting of hundreds of atoms.

Linear-scaling electronic structure methods, such as linear-scaling density functional theory, are valuable tools for ab initio simulations of large systems. These methods assume that the electronic structure has a short-range nature and achieve linear-scaling by truncating elements beyond a given cutoff radius or below a given threshold[62]. In contrast, our approach does not impose any localization constraints - selected features span a wide range of distances and contributions. The descriptor reduction procedure allows us to find the right low-dimensional embedding of the high-dimensional PES. Furthermore, the linear-scaling electronic structure methods are less accurate than the original $\mathcal{O}(N^3)$ approaches by design. The models trained with reduced descriptors provide predictions with equal or better accuracy than the original models since the deprecated features, as we have shown, constitute noise in the model.

## Discussion

Efficient modeling of large molecules requires descriptors of low dimensionality that include relevant features for a particular prediction task. Our results show that beyond increasing the efficiency, such descriptors improve the accuracy of ML models compared to those constructed with default global or local descriptors. This is the consequence of simplifying the interaction patterns which should be learned by ML models in the reduced descriptor spaces. The resulting MLFFs allow long-time molecular dynamic simulations demonstrating stable behavior in the regions of the PES represented in the training sets.

A detailed analysis of the non-local descriptor features relevant for accurate energy/force predictions shows non-trivial patterns. These patterns are related to the molecular structure and composition, balancing the strength of the interactions associated with the descriptor features and statistical information about atomic fluctuations these features provide. In particular, we show that the descriptor features related to interatomic distances as large as 15 Å can play an essential role in describing non-local interactions. Our examples cover units of all four major classes of biomolecules and supramolecules, making the conclusions general for a broad range of (bio)chemical systems.

The key outcome of the proposed descriptor reduction scheme is the linear scaling of the resulting global descriptors with the number of atoms. We found that global descriptors for large molecules are over-defined and equally accurate models can be constructed with just a handful of long-range features that describe collective long-range

interactions. This behavior seems to be general for large molecular systems, provided that reliable reference data is available.

Overall, our work makes substantial breakthroughs in the broad domain of machine learning force fields. These breakthroughs include (i) demonstrating the potential for linear scaling in global MLFFs for large systems, (ii) analyzing the non-local interatomic features that contribute to accurate predictions, and (iii) demonstrating the accuracy, efficiency, and stability of reduced models in long time-scale molecular dynamics simulations. As such, this is a critical step for building accurate, fast, and easy-to-train MLFFs for systems with hundreds of atoms without sacrificing collective non-local interactions.

## Methods
### Interaction heatmaps
We use corresponding GDML (ver. 0.4.11) models trained and validated on 1000 different configurations to calculate interaction heatmaps (Fig. 1b, d and Fig. 3). Heatmaps consist of pairwise contributions, $F_l^k$, averaged over many configurations from the dataset (3000 configurations for the Ac-Ala3-NHMe and AT-AT; 1000 configurations for the buckyball catcher).

The trained GDML force field estimator collects the contributions of the $3N$ partial derivatives ($N$ - number of atoms) of all $M$ training points to compile the prediction:

$$\mathbf{F}(\mathbf{x}) = \sum_{i=1}^{M} \sum_{j=1}^{3N} (\boldsymbol{\alpha}_i)_j \frac{\partial}{\partial x_j} \nabla \kappa(\mathbf{x}, \mathbf{x}_i), \tag{1}$$

where $\mathbf{F}(\mathbf{x})$ is a vector containing the $3N$ forces predicted for molecular geometry $\mathbf{x}$.

A partial evaluation of this sum yields the contribution of a single atom $k$ to the force prediction on all atoms (atom indices start from 0):

$$\mathbf{F}^k(\mathbf{x}) = \sum_{i=1}^{M} \sum_{j=3k+1}^{3k+3} (\boldsymbol{\alpha}_i)_j \frac{\partial}{\partial x_j} \nabla \kappa(\mathbf{x}, \mathbf{x}_i). \tag{2}$$

To obtain the contribution of atom $k$ to the force prediction on atom $l$, $F_l^k$, we compute the norm of the force components of vector $\mathbf{F}^k(\mathbf{x})$ that correspond to an atom $l$:

$$F_l^k = \left\{ \sum_{s=1}^{3} \left( \mathbf{F}^k(\mathbf{x})_{3l+s} \right)^2 \right\}^{1/2}. \tag{3}$$

### Descriptor reduction
The procedure starts from a pre-trained kernel-based ML model ($\text{ML}_{original}$), with a default global (containing all $n$ features) descriptor $\mathbf{x}$. Importantly, we do not require a highly accurate and thus computationally expensive ML model at this stage (see Methods for further details).

The significance of the $n$-th feature in the descriptor is obtained by comparing the prediction results on a subset of test configurations between the full $\text{ML}_{original}$ and the $\text{ML}_{original}$ with the $n$-th feature set to zero for all configurations ($\text{ML}_{mask}^n$). Thus, we assume separability between the features in the descriptor, but this does not imply their independence. Therefore, more advanced and computationally expensive reduction techniques can also be applied[63,64].

This procedure is performed separately for all features in the descriptor. All other parameters of the $\text{ML}_{original}$ model remain unchanged when obtaining the $\text{ML}_{mask}^n$ predictions. Therefore, the only difference between the models is in the definition of similarity

**Table 1 | Settings of the MD simulations of the datasets used in the work**

| Molecule | Basis set | Temp. | Thermostat | Coef. |
|---|---|---|---|---|
| Ac-Ala3-NHMe | tight | 500 | Global Langevin | 2 |
| AT-AT | tight | 500 | Global Langevin | 2 |
| Buckyball catcher | light | 400 | Nosé-Hoover | 1700 |
| Lactose | light | 500 | Nosé-Hoover | 1700 |
| Palmitic acid | light | 500 | Nosé-Hoover | 1700 |

Temperature is given in *K*. PBE stands for the Perdew-Burke-Ernzerhof functional and MBD stands for many-body dispersion[74]. Coefficient refers to the friction coefficient (in fs) for the global Langevin thermostat, and to the effective mass (in cm$^{-1}$) for the Nosé-Hoover thermostat.

between system states. The loss function

$$L_n = \sum_{i=1}^{N} \left( ML_{original}(\mathbf{x}_i) - ML_{mask}^n(\mathbf{x}_i) \right)^2, \tag{4}$$

where $N$ is the number of test configurations, serves as a measure of the importance of a particular feature $n$ in the descriptor. The descriptor features where the loss $L_n$ is the smallest are the least important for the model and can be removed from the descriptor. As soon as our analysis is performed on a representative subset of configurations, we ensure that we preserve all the descriptor features relevant for modeling the given PES. However, setting a threshold under which one can consider a feature as irrelevant is not trivial. The values of $L_n$ depend on (i) the predicted property, (ii) the system(s) for which the model is trained, and iii) the reference data used for training. In the study, we consider every 10th percentile of all $L_n$ values (10th to 90th). As a final step, a new ML model is trained and tested after removing from the default descriptor all the features whose corresponding $L_n$ are below a selected percentile.

### Reference datasets
Molecular dynamics (MD) simulations at PBE+MBD level of theory with a step size of 1 fs were used to construct the reference datasets. Table 1 includes all other relevant information about the datasets. Calculations were done either with i-PI[65] wrapped with FHI-aims code[66] to compute forces and energies or with FHI-aims code alone.

### Computational details for ML models
The ML models were built with GDML[1,2] and GAPs[8] with the SOAP representation[30]. GDML models were trained using a numerical solver with an initial value of 70 inducing points. All models were validated using 1000 configurations and hyperparameter search $\sigma$ was performed individually for each system and training size to ensure optimal model selection (from 10 to 1000). No symmetries were considered in the models for a fair comparison between the default descriptor and those with a reduced size. GAP/SOAP models were trained using 12 radial and 6 angular functions for the descriptor. The cutoff radius was set to 5 Å. Parameter $\delta$ was set to 0.25, the atom $\sigma$ was set to 0.3, and the default $\sigma$s for energy and forces were set to 0.001 and 0.1, respectively. These calculations were performed with the QUIP program package[67] through the quippy python interface[68].

### Molecular dynamics simulations
External-force DFT calculations were performed using the FHI-aims electronic structure software in combination with the external force option in the Atomic Simulation Environment package[69]. We used the PBE and PBE+MBD[70,71] level of theory with the *intermediate* basis set. Trajectories were generated with a resolution of 0.5 fs and sampled at 300 K using a Langevin thermostat with a friction coefficient of $1 \times 10^{-3}$. 

To evaluate the performance of machine learning models, we utilized both the ML$_{Global}$ and ML$_{R0.6}$ models trained on 5000

configurations with the same settings. To ensure stability during 17 parallel simulation dynamics, each averagin around 3 ns (total time of 50 ns), we utilized a timestep of 0.3 fs and a Langevin thermostat with a $1 \times 10^{-4}$ friction coefficient. The first configuration of Ac-Ala3-NHMe from the dataset was used as a starting configuration in all calculations.

### Important considerations of the descriptor dimensionality reduction approach
We discuss some considerations that are important when applying the descriptor reduction approach. The results presented here were obtained with the GDML method[1,2] using its default descriptor (i.e., all inverse pairwise distances), but any other method or descriptor could have been used as well.

1. The result of the dimensionality reduction approach is weakly dependent on the training set used for training the ML$_{original}$ model. For example, we trained 5 different models for aspirin (210 features in the descriptor) with 500 training points and used 5 different subset of 3000 configurations (one for each model) to select the root mean square errors (RMSEs) under the 15th percentile. From the 32 removed features, 24 of them (75%) were removed with all 5 aspirin models. This means that when applying the dimensionality reduction one only needs to ensure that the training set is representative of the dataset.

2. The result of the dimensionality reduction approach is independent of the training set size used for training the ML$_{original}$ model. For instance, we trained 3 different models for alanine tetrapeptide (Ac-Ala3-NHMe; 861 features in the descriptor) with 100, 200, and 500 configurations and used a subset of 3000 configurations (the same for all models) to select the RMSEs under the 65th percentile. From the 559 removed features, 443 of them (~79%) were removed with all three models. This means that one does not need to start with a very accurate (probably expensive) initial model.

3. The result of the dimensionality reduction approach is independent of the size of the subset used for computing the ML$_{original}$ and the ML$_{masked}^n$ predictions. As an example, we trained a model for an adenine-thiamine DNA base-pair dimer (AT-AT; 1770 features in the descriptor) with 100 training points and used different subsets with 10, 100, and 1000 configurations for selecting the RMSEs under the 60th percentile. From the 1062 removed features, 1030 of them (~97%) were removed with all subsets. This is advantageous because one can efficiently assess the importance of all features in the representation of a molecule, even if the descriptor has thousands of features due to the size of the molecule.

4. The result of the dimensionality reduction approach is weakly dependent on the regularization coefficient, as long as the ML$_{original}$ model is still accurate. For example, we trained three models with 1000 training points for Ac-Ala3-NHMe with 3 regularization coefficients ($10^{-8}$, $10^{-10}$, $10^{-12}$). 81–95% of the selected features are the same between the three models; 93% same for ML$_{R0.1}$.

The reason why the overlap between removed features in the examples of points 1 and 2 (<80%) is not as high as in the one of point 3 (~97%) is simple. Most of the features that are not removed by all models (e.g., the remaining 8 features in each aspirin model discussed in point 1) involve a hydrogen atom. Hydrogen atoms are the ones that fluctuate the most in MD simulations, which are the origin of the datasets. Thus, it is not surprising that the relevance of a given feature involving a hydrogen atom varies between different ML models (trained on different sets) without affecting the reliability of the resulting force field.

### Data availability
Datasets for Ac-Ala3-NHMe, AT-AT, and the buckyball catcher are now part of the MD22 dataset[57], available at www.sgdml.org.

## Code availability

The (s)GDML code used in this work is available at https://github.com/stefanch/sGDML. The code for obtaining the reduced descriptors and the modified (s)GDML routines are available at https://doi.org/10.5281/zenodo.7876825[73].

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

## Acknowledgements
A.K., V.V.G, I.P, and A.T. acknowledge financial support from the Lux-embourg National Research Fund (FNR) (AFR Ph.D. Grant 15720828 and Grant C19/MS/13718694/QML-FLEX) and the European Research Council (ERC) Consolidator Grant BeStMo. S.C. acknowledges support from the Federal Ministry of Education and Research (BMBF) for BIFOLD (01IS18037A). The results presented in this work have been obtained using the HPC facilities of the University of Luxembourg[72]. We thank IPAM for its warm hospitality and inspiration while finishing the manuscript.

## Author contributions
A.K. and V.V.G. contributed equally to this work. V.V.G. and I.P. conceived the proposed approach. V.V.G. and S.C. implemented the methodology in GDML. A.K., V.V.G, I.P., and A.T. designed the analyses. A.K. and V.V.G. carried out the reference quantum chemical calculations and trained the ML models. A.K. performed the major part of the analyses. A.K. created the figures with the help from other authors. A.K., V.V.G, I.P., and A.T. wrote the paper. All authors discussed the results and commented on the manuscript. I.P. and A.T. supervised and guided the project.

## Competing interests
The authors declare no competing interests.
