## [Peer Review File · Nature Communications]

Efficient Interatomic Descriptors for Accurate Machine Learning Force Fields of Extended MoleculesREVIEWER COMMENTS

Reviewer #1 (Remarks to the Author):

Review of Tkatchenko et al.

This manuscript presents a step towards linear scaling kernel-based machine learning methods for representing and evaluating energy functions for molecules. The work is technically sound and focuses primarily on the performance of the strategy to reduce the size of the descriptor.

Detailed comments:

1. The ansatz is reminiscent of approaches in electronic structure theory to arrive at $O(N)$ scaling. The authors may want to mention this. Looked at the findings from this perspective, the reported results are not entirely surprising.
2. Is the model performance in Figure 1E for one particular structure? It is relevant to report such a graph for an ensemble of structures together with error bars or error distributions. Also, a comparison of the RMSD for "in sample" and "out of sample" structures with the size of the descriptor and its performance on energies/forces will be required.
3. It appears that conventionally used descriptors are overdetermined and their use leads to "noise" in the trained model which leads to decreased performance/accuracy. Can the authors clarify?
4. Related to point 2 above: how well do the reduced descriptors extrapolate for out-of sample structures? For example, when training on "compact structures" how does the model perform on "extended structures" of peptides?
5. It appears that the way how the size of the descriptor is reduced assumes separability between the features, see "Descriptor reduction" on p. 9. Is this true and is this a justified assumption?
6. The last sentence "We should note that..." requires citations.
7. Do citations 53 and 54 have arxiv numbers? They should be given.

In summary, the work is more on the technical side and does not offer big "surprises". For better delineating the validity and applicability of the approach it should be clarified how structurally diverse the out-of-sample data set is and how this affects model performance.

Reviewer #2 (Remarks to the Author):

This manuscript reports an interesting analysis of ML models for the interatomic interactions in molecules based on global descriptors such as the Coulomb matrix. While accurate, a drawback of these descriptors is that they scale quadratically with system size, limiting their application to large

molecules. The authors show that this limitation can be overcome by retaining only the essential long-range features of the global descriptors. These features can be identified automatically with a prescription provided in the manuscript, without recourse to a specific physics model. The resulting reduced global ML models are found to scale linearly with system size in a relatively broad set of molecules, representative of rather flexible building units found in biomolecules. The finding that linearly scaling reduced global models are possible, is novel and important. The manuscript is well written and easy to read. In order to recommend publication, however, I wish the authors should address the following issues.

1. While linearly scaling global models are important, still they need to be trained with DFT calculations that typically scale like the third or the fourth power of the number of electrons, implying that learning the essential long-range features of an ML model will be limited to relatively small systems. How do the authors intend to address this issue? There are, of course, linear scaling DFT codes out there, but their accuracy/applicability is limited compared to general purpose codes, and the cost of these calculations for the large number of configurations necessary for training can still be prohibitive.

2. Linear scaling is not demonstrated "in general" as is often stated improperly in the manuscript, but only up to the sizes of the test molecular systems, the largest of which, the buckyball catcher, is made of 148 atoms. Absent a physical model for the long-range interactions and just relying on ML, how can the results of the manuscript be generalized to larger molecular systems? The authors mention macromolecules as key targets of their approach, but they only consider some simple molecular building blocks. It would be interesting if they could show that the linear scaling behavior found for a building block could be generalized to flexible polymer fragments, made by several building blocks. The potential for generalization of the reported results needs to be better specified/quantified. What if long-range charge transfer effects are present?

3. The authors mention, as a possible future outlook, switching from atom centered non-local features to more efficient and general collective coordinates that would provide effective interaction centers such as those employed in the TIP4P model of water and similar empirical force fields. However, models using more general collective coordinates (not purely atom-centered) have already been proposed in the context of ML potentials trained on DFT data (see, e.g., L. Zhang et al., JCP 156, 124107 (2022)). These models are derived from the physics of the long-range interactions. Would it be possible to infer a sound physical model from the analysis of the non-local interaction patterns observed in the manuscript? This could provide a missing link needed to properly generalize the reduced global models to large (and even infinite) system sizes.

4. Calculations reported in the manuscript include many-body dispersion (MBD), an effect beyond semi-local DFT. How much does MBD contribute to the non-local features of the global descriptor, would these features be present also in a pure PBE model?

5. Fig. S5 of the supporting information shows force error distributions with different models. While it appears from the data that MLopt and MLglobal models are superior to MLsoap, it is not evident that MLopt is superior to MLglobal in the tail of the distributions. What is the statistical significance of the oscillatory behavior of MLglobal in Fig S3C?

Reviewer #3 (Remarks to the Author):

Noteworthy results:

In this manuscript, Kabylda et al. present an approach for reducing the number of descriptors needed in machine learning force fields (MLFFs) and simultaneously reducing the computational cost to scale linearly with the number of descriptors. Importantly, the scaling prefactor in the current method is

nearly the same as the more costly approach, such that a true speedup results. The authors demonstrate their method on a variety of model systems of broad interest.

Significance to the field and related fields:

This work will significantly impact the development of machine learning force fields and their use in broad variety of fields.

Does the work support the conclusions and claims?:

The conclusions are indeed supported by the work presented in the paper.

There are no flaws and the methodology is sound. There are indeed enough details to reproduce the work.

This is an excellent manuscript that should be published after minor revisions.

My very minor comments are listed below.

1. On line 90, the authors suggest that MLFFs should not involve defining a characteristic lengthscale, but then later define a lengthscale to distinguish short range interactions from long range interactions. Can the authors reconcile these differences?

2. The general idea of separately focusing on short range and long range interactions to enable simpler modeling of long range interactions without loss in accuracy seems to be complementary to a couple recent papers on MLFFs: Niblett, Galib, and Limmer, J Chem Phys 2021 (NGL), <https://doi.org/10.1063/5.0067565>, and Gao and Remsing, Nature Commun. 2022 (GR), <https://doi.org/10.1038/s41467-022-29243-2>. Although these papers focus on using established descriptors to separately model short and long range interactions, and this manuscript focuses on developing reduced descriptors, the general idea of leveraging fundamental differences between short and long range interactions to simplify the problem seem to be similar in spirit. For example, NGL were able to use fixed point charges to accurately model long range electrostatics in water with MLFFs trained on ab initio data. In addition, GR found that simpler neural networks could be used to capture long range physics with a proper splitting of short and long range interactions. The method developed in current manuscript could very nicely complement these approaches. If the authors agree, it may be useful to connect to approaches like these to broaden the relevance of the work.

3. On lines 441-442, the authors state that their "resulting MLFFs allow long-time molecular dynamic simulations..." but earlier they state that the MD was only stable for a relatively short time, ~300 ps (lines 335-336). Can the authors please resolve this contradiction?

4. Line 300, delete "provide"

5. Line 424, delete one "the"

Reviewer 1

Review of Tkatchenko et al.

This manuscript presents a step towards linear scaling kernel-based machine learning methods for representing and evaluating energy functions for molecules. The work is technically sound and focuses primarily on the performance of the strategy to reduce the size of the descriptor.

We thank the referee for carefully reading our manuscript and his/her thorough comments.

Detailed comments:

1. The ansatz is reminiscent of approaches in electronic structure theory to arrive at $O(N)$ scaling. The authors may want to mention this. Looked at the findings from this perspective, the reported results are not entirely surprising.

The approximations employed for constructing linear-scaling electronic structure methods might look akin to our descriptor reduction approach. However, there are essential differences between both ansätze.

In $O(N)$ electronic-structure methods, localization is imposed by dividing the entire system into sub-systems and then defining electronic degrees of freedom that are localized within them (i.e. notion of locality) [1, 2]. Similar ideas make the basis for (linear scaling) local descriptors/models, such as the GAP/SOAP model against which we compare our reduced model.

This assumption is justified in some cases, such as small/rigid molecules and materials. For instance, when we consider a relatively rigid lactose molecule, a localized description of interactions proves to be most suitable; this descriptor can be equally obtained with introducing a cut-off radius. However, locality assumption is less justified for large and flexible molecules. Therefore, we emphasize the need for "MLFFs without employing strictly predefined functional forms for interactions or imposing characteristic length scale". Consequently, we do not impose any localization constraints in our approach - selected features span a wide range of distances and contributions for these large and flexible systems. Descriptor reduction procedure allows us to identify the right low-dimensional embedding of the high-dimensional PES.

Furthermore, linear-scaling electronic structure methods provide less accurate predictions than the original $O(N^3)$ approaches by construction. The models trained with reduced descriptors provide predictions with equal or better accuracy than the original models since the deprecated features, as we have shown with additional analysis (please see reply to point 2 of Reviewer 2), constitute noise in the model.

We have added discussion on these important points in the revised manuscript (lines 485-502).

[1] G. Galli, *Curr. Opin. Solid State Mater. Sci.* 1, 864 (1996)

[2] D.R. Bowler and T. Miyazaki, *Rep. Prog. Phys.*, 75, 036503 (2012)

2. Is the model performance in Figure 1E for one particular structure? It is relevant to report such a graph for an ensemble of structures together with error bars or error distributions. Also, a comparison of the RMSD for "in sample" and "out of sample" structures with the size of the descriptor and its performance on energies/forces will be required.

The performance of the models (Force / Energy RMSE) in Figure 1E and Figure 2 were obtained for an ensemble of structures in the test set. For instance, the curves shown in Figure 1E are the result of the prediction of 83109 configurations. We have modified the caption of Fig. 1 to make this point clear.

The structures in the training, validation, and test sets do not overlap. However, they all preserve the original energy distribution of the entire dataset. This is achieved by employing sGDML's default training/validation/test set selection scheme. In this scheme, sGDML draws a sample from the original energy distribution. The distribution is estimated from a histogram where the bin size is determined using the Freedman-Diaconis rule. This rule is designed to minimize the difference between the area under the empirical probability distribution and the area under the theoretical probability distribution. The reduced histogram is then constructed by uniform sampling in each bin. All bins are populated with at least one

sample in the reduced histogram, even for small set sizes.

Regularization is used to avoid overfitting the training data. Therefore, the Force / Energy RMSEs for train (“in sample” data) and test sets (“out of sample” data) are similar. The distribution of errors in the test set was plotted in the original manuscript (Fig. S4, Fig. S1 in the revised manuscript).

Your question motivated us to test whether the regularization coefficient affects the reduction procedure. The selected features are weakly dependent on it, as long as the model is still accurate. We have clarified this in the revised manuscript (lines 168-172) and in the revised supplementary file (lines 34-38).

3. It appears that conventionally used descriptors are overdetermined and their use leads to “noise” in the trained model which leads to decreased performance/accuracy. Can the authors clarify?

In line 133 of the original manuscript (line 141 of the revised manuscript), we briefly referred to this by saying: “For molecules containing just a few dozen of atoms, such [global] descriptors are, in fact, substantially over-defined.” Over-defined here means the dimensionality of the global descriptors ($N*(N-1)/2$) is much larger than $3N-6$ (N is the number of atoms), which is needed to define the system configuration uniquely. One of the main consequences of this is related to the statement of the reviewer. Namely, an oversized descriptor will span a much larger space than what is effectively needed, making ML models harder to optimize and compromising their performance/accuracy. Our approach solves the problem of identifying the relevant degrees of freedom in descriptor space, in a consistent manner across the whole training set.

We have clarified this point and provided further analysis that shows that there is a “noise” in the global models, that is ameliorated in the reduced models in the revised manuscript (lines 147-149 and 380-390, as well as Fig. S4).

4. Related to point 2 above: how well do the reduced descriptors extrapolate for out-of sample structures? For example, when training on “compact structures” how does the model perform on “extended structures” of peptides?

We have compared the transferability of the global and reduced models for Ac-Ala3-NHMe, AT-AT, and the Buckyball Catcher in the scenario suggested by the reviewer. For all three cases, we took the 90th percentile of a dataset with respect to the total energy and trained global and reduced models based on it. Then, the remaining structures (“extended” in the case of the tetrapeptide) were used for testing the performance. The reduced models show an improvement in prediction of the high-energy outliers (see Table below). Therefore, the reduced model (by having fewer features) is less likely to overfit, meaning it can more accurately predict outcomes on unseen outlier data.

Molecule	Model	Energy RMSE [kcal mol ⁻¹]	Force RMSE [in kcal (mol Å) ⁻¹]
Ac-Ala3-NHMe	ML _{Global}	2.05	2.97
	ML _{R0.4}	1.84	2.80
AT-AT	ML _{Global}	6.56	2.51
	ML _{R0.4}	4.56	2.46
Buckyball Catcher	ML _{Global}	1.60	1.98
	ML _{R0.2}	1.46	1.61

5. It appears that the way how the size of the descriptor is reduced assumes separability between the features, see “Descriptor reduction” on p. 9. Is this true and is this a justified assumption?

We assume separability between the features during the descriptor reduction procedure, but this does not imply their independence since features and interactions they describe are intrinsically coupled. Therefore, in principle, more advanced and computationally expensive reduction techniques can be applied that would explore more combinations of features (not masking them one-by-one) [Refs. 63 and 64 in the revised manuscript]. Nevertheless, we think they are impractical for such high-dimensional descriptors (of large molecules), and the resulting gain would be negligible.

We have added this remark in the “Descriptor reduction” section of our revised manuscript (lines 583-587).

6. The last sentence "We should note that..." requires citations.

The citations have been added in the revised manuscript (Refs. 63 and 64).

7. Do citations 53 and 54 have arxiv numbers? They should be given.

The citations 53 and 54 have been updated in the revised manuscript (Refs. 57 and 58).

In summary, the work is more on the technical side and does not offer big "surprises". For better delineating the validity and applicability of the approach it should be clarified how structurally diverse the out-of-sample data set is and how this affects model performance.

We are grateful to the reviewer for his/her comments. We believe that in this point-by-point response and in the revised manuscript we have highlighted and properly delimited the validity and applicability of what we propose. Moreover, we hope that the revisions we have made demonstrate not only a technical soundness, but also a degree of novelty in our findings.

Reviewer 2

This manuscript reports an interesting analysis of ML models for the interatomic interactions in molecules based on global descriptors such as the Coulomb matrix. While accurate, a drawback of these descriptors is that they scale quadratically with system size, limiting their application to large molecules. The authors show that this limitation can be overcome by retaining only the essential long-range features of the global descriptors. These features can be identified automatically with a prescription provided in the manuscript, without recourse to a specific physics model. The resulting reduced global ML models are found to scale linearly with system size in a relatively broad set of molecules, representative of rather flexible building units found in biomolecules. The finding that linearly scaling reduced global models are possible, is novel and important. The manuscript is well written and easy to read. In order to recommend publication, however, I wish the authors should address the following issues.

We thank the reviewer for highlighting the novelty and importance of the work and for valuable comments/suggestions.

1. While linearly scaling global models are important, they still need to be trained with DFT calculations that typically scale like the third or fourth power of the number of electrons, implying that learning the essential long-range features of an ML model will be limited to relatively small systems. How do the authors intend to address this issue? There are, of course, linear scaling DFT codes out there, but their accuracy/applicability is limited compared to general purpose codes, and the cost of these calculations for the large number of configurations necessary for training can still be prohibitive.

ML Force Fields can only provide reliable predictions for structures similar to those sampled in the training set. Therefore, any ML approach that aims to reconstruct a PES with DFT accuracy must be trained on DFT reference data. This is true for all state-of-the-art ML models/architectures and is not a deficiency of our method in which we address the quadratic scaling of global ML models. Many efforts are put into developing ML methods that are more data-efficient, reducing the amount of DFT calculations needed for training.

In this regard, the linearly scaling global models presented in our work represent an important step toward more data-efficient ML models (i.e. ML models that require fewer DFT calculations). This can be concluded from the performance of our reduced models with respect to the original global models. Furthermore, model constructed with a minimal amount of features should, in principle, be more transferable. We have shown first results in this direction in response to point 4 of Reviewer 1. The reduced model is more likely to capture the essential characteristics of the problem and less likely to overfit the data. This makes it easier to generalize the model to unseen data.

2. Linear scaling is not demonstrated “in general” as is often stated improperly in the manuscript, but only up to the sizes of the test molecular systems, the largest of which, the buckyball catcher, is made of 148 atoms. Absent a physical model for the long-range interactions and just relying on ML, how can the results of the manuscript be generalized to larger molecular systems? The authors mention macromolecules as key targets of their approach, but they only consider some simple molecular building blocks. It would be interesting if they could show that the linear scaling behavior found for a building block could be generalized to flexible polymer fragments, made by several building blocks. The potential for generalization of the reported results needs to be better specified/quantified. What if long-range charge transfer effects are present?

We agree with the reviewer that the generalization of our linearly scaling global descriptor to larger molecules and systems with a more complex interplay of non-local interactions (e.g. long-range charge transfer effects) is important to assess.

Therefore, we have done a further analysis to better understand why exactly reduced models work better and whether this can be generalized to larger systems. We find that contribution of particular features in the global model can range from linear to stochastic with respect to the interatomic distance (Fig. S4A). The proportion of stochastic features increases with the size of the systems and the size of training set (Fig. S4B). In the reduced models after retraining most of the selected features have a high coefficient of determination, R^2 (Fig. S4C). Thus, the stochastic features constitute a noise in the global models.

Contribution of “linear” features to the force prediction decrease quadratically with distance (slope ≈ -2), suggesting the prominence of Coulombic contributions to the interatomic forces. There is a deviation from the linearity for some features in the long-range regime; therefore, other power-laws are also learned by the model.

Thus, we additionally confirm that the global descriptors for large molecules are over-defined. Removing this noise with the proposed reduction procedure leads to a more or equally accurate models, where a few of long-range features describe collective long-range interactions. This can be envisioned to be general for any system and used to train more effective ML models, provided that we have reliable reference calculations.

We also note that the molecular building blocks on which we test our approach are one of the largest molecules simulated with ML potentials. Some of them are now included in the recently published MD22 benchmark dataset that presents new challenges for ML models with respect to system size, flexibility, and degree of non-locality [3].

The corresponding discussions have been added in the revised manuscript (lines 380-390 and 537-544).

[3] S. Chmiela et al. arXiv:2209.14865 [physics.chem-ph] (2022)

3. The authors mention, as a possible future outlook, switching from atom centered non-local features to more efficient and general collective coordinates that would provide effective interaction centers such as those employed in the TIP4P model of water and similar empirical force fields. However, models using more general collective coordinates (not purely atom-centered) have already been proposed in the context of ML potentials trained on DFT data (see, e.g., L. Zhang et al., JCP 156, 124107 (2022)). These models are derived from the physics of the long-range interactions. Would it be possible to infer a sound physical model from the analysis of the non-local interaction patterns observed in the manuscript? This could provide a missing link needed to properly generalize the reduced global models to large (and even infinite) system sizes.

We thank the reviewer for bringing our attention to the referenced article. Indeed, long-range effects can play an important role, limiting the predictive power of local models in nanoscale systems. Therefore, some recent MLFF models have integrated correction terms to account for certain long-range effects (e.g. electrostatics), but long-range electron correlation effects are still not well characterized. It is evident that the field of MLFF combined with physical interaction models is rapidly growing and developing, but a definitive solution to these challenges has not yet been found.

We emphasize that length scales cannot be separated in an additive fashion due to the “non-additivity” of short- and long-range effects. Therefore, global and proposed reduced models describe all interaction scales on an equal footing and do not require separate augmentations to describe long-range interactions. Corresponding quadratic scaling of global models provides a severe computational constraint and has therefore slowed the development of global models in recent years [3]. The presented advancement is intended to push the boundaries of what is possible with current global models. Our analysis offers ways to build future-generation ML models, such as multi-scale models that are based on effective interaction centers, which can be used for complex molecules, where obtaining a coarse-grained representation is not as straightforward as with the TIP4P model or Wannier centroids for water molecules referenced by the reviewer.

The corresponding discussions have been added to the text (lines 81-89 and 221-224).

[3] S. Chmiela et al. arXiv:2209.14865 [physics.chem-ph] (2022)

4. Calculations reported in the manuscript include many-body dispersion (MBD), an effect beyond semi-local DFT. How much does MBD contribute to the non-local features of the global descriptor, would these features be present also in a pure PBE model?

We have evaluated the importance of MBD contributions by comparing two models and the resulting features from two molecules: Ac-Ala3-NHMe and Buckyball catcher. The first model was trained on the initial dataset computed at the PBE+MBD level of theory, while the second model was trained using the new dataset, where energies and forces were recalculated at the PBE level of theory for the geometries from the initial

PBE+MBD dataset.

We find that long-range descriptor features are also kept in PBE calculations (without MBD). This is because PBE functional includes long-range electrostatics and polarization despite the fact that exchange-correlation is semi-local. However, when we account for MBD, 1-22% of features can change depending on the degree of reduction. For example, our $ML_{R0.9}$ model differs by 22% and the most reduced $ML_{R0.1}$ model differs by only 1% since most long-range effects are omitted in this case. The difference for the optimal models $ML_{R0.4-0.6}$ is around 5%.

Our findings are consistent with the fact that MBD contribution to the energy is relatively small compared to the PBE energy. Nevertheless, MBD energies can greatly influence the dynamics of molecules. For example, MBD is essential for accurately evaluating the stability of aspirin polymorphs [4], standing molecules on surfaces [5], and interlayer sliding of 2D materials [6]. The importance of long-range vdW interactions is expected to be even greater for large and flexible molecules. Therefore, it is essential to have PBE+MBD calculations to produce a reliable dataset in the first place.

We have added these results in the main text (lines 408-426).

[4] A. M. Reilly and A. Tkatchenko, Phys. Rev. Lett. 113, 055701 (2014)

[5] M. Knol et al., Sci. Adv. 7, eabj9751 (2021)

[6] W. Gao and A. Tkatchenko, Phys. Rev. Lett. 114, 096101 (2015)

5. Fig. S5 of the supporting information shows force error distributions with different models. While it appears from the data that ML_{opt} and ML_{global} models are superior to ML_{soap} , it is not evident that ML_{opt} is superior to ML_{global} in the tail of the distributions. What is the statistical significance of the oscillatory behavior of ML_{global} in Fig S3C?

The oscillatory behavior was the consequence of an issue of the kernel density estimate (KDE) at the tails of the distributions. To rectify this, we now have plotted the step histograms directly. The superiority of ML_{opt} over ML_{global} models is now more evident. The figure has been updated in the revised manuscript (Fig. S1).

Reviewer 3

Noteworthy results: In this manuscript, Kabylda et al. present an approach for reducing the number of descriptors needed in machine learning force fields (MLFFs) and simultaneously reducing the computational cost to scale linearly with the number of descriptors. Importantly, the scaling prefactor in the current method is nearly the same as the more costly approach, such that a true speedup results. The authors demonstrate their method on a variety of model systems of broad interest.

Significance to the field and related fields: This work will significantly impact the development of machine learning force fields and their use in broad variety of fields.

Does the work support the conclusions and claims?: The conclusions are indeed supported by the work presented in the paper.

There are no flaws and the methodology is sound. There are indeed enough details to reproduce the work.

This is an excellent manuscript that should be published after minor revisions. My very minor comments are listed below.

We thank the referee for providing feedback on our manuscript and for highlighting the significance of our work.

1. On line 90, the authors suggest that MLFFs should not involve defining a characteristic lengthscale, but then later define a lengthscale to distinguish short range interactions from long range interactions. Can the authors reconcile these differences?

The model and the selection scheme do not rely on any characteristic lengthscale. In fact, we show that the reduced descriptors have features that correspond to interatomic distances that are on average even as far as 15 Å. The distinction between short- and long-range interactions, for instance in Figure 4B, only serves to show the effect of our descriptor reduction approach on the different types of interactions as conventionally defined when imposing lengthscales, i.e. distances shorter than 5 Å correspond to short-range interactions and distances larger than that length correspond to long-range ones.

We have clarified this point in the revised manuscript (lines 392-395).

2. The general idea of separately focusing on short range and long range interactions to enable simpler modeling of long range interactions without loss in accuracy seems to be complementary to a couple recent papers on MLFFs: Niblett, Galib, and Limmer, J Chem Phys 2021 (NGL), <https://doi.org/10.1063/5.0067565>, and Gao and Remsing, Nature Commun. 2022 (GR), <https://doi.org/10.1038/s41467-022-29243-2>. Although these papers focus on using established descriptors to separately model short and long range interactions, and this manuscript focuses on developing reduced descriptors, the general idea of leveraging fundamental differences between short and long range interactions to simplify the problem seem to be similar in spirit. For example, NGL were able to use fixed point charges to accurately model long range electrostatics in water with MLFFs trained on ab initio data. In addition, GR found that simpler neural networks could be used to capture long range physics with a proper splitting of short and long range interactions. The method developed in current manuscript could very nicely complement these approaches. If the authors agree, it may be useful to connect to approaches like these to broaden the relevance of the work.

We thank the referee for the suggestion, which we have incorporated into the revised manuscript to better delimit the scope of the work (lines 81-89).

We would like to remark that the distinctions we make between short- and long-range features along the manuscript are only employed to analyze the features in our reduced descriptors from the perspective of conventional definitions of locality and non-locality. Original global (s)GDML models and the reduced models proposed in the present work do not rely on any separation between short- and long-range interactions. Both models treat all types of interaction on an equal footing (please also see the reply to point 3 of Reviewer 2).

3. On lines 441-442, the authors state that their “resulting MLFFs allow long-time molecular dynamic simulations. . .” but earlier they state that the MD was only stable for a relatively short time, 300 ps (lines

335-336). Can the authors please resolve this contradiction?

Stability is an issue for many Machine Learning Force Fields, as recently shown in Ref. [7] (published after submission of this manuscript). For small organic molecules from MD17 dataset, authors perform 300 ps simulations to assess the stability of various ML models [7]. In our case, we have also tested the stability in the simulations for at least 300 ps. Further stability depends on the accuracy of the underlying original model. The buckyball catcher would be stable for a much longer time since the model needs less data for the relatively rigid molecular complex. In the case of the tetrapeptide and DNA base pairs, more accurate original models and/or active learning are needed as we discussed in the original manuscript.

We have clarified this contradiction in the revised manuscript (lines 348-350).

[7] X. Fu et al., arXiv:2210.07237 [physics.comp-ph] (2022)

4. Line 300, delete “provide”

5. Line 424, delete one “the”

We have corrected mentioned typos in the revised manuscript.

REVIEWER COMMENTS

Reviewer #1 (Remarks to the Author):

Review of Tkatchenko et al.

The authors have made several modifications and provided clarifications on points raised by the reviewers. A few points still remain to be addressed, nevertheless.

Detailed comments:

1. What do the the authors refer to when writing "However, they all preserve the original energy distribution..", that the $P(E)$ for the training, validation and test sets are similar/identical?

2. It is still not clear whether features for individual structures for a database of molecules can be used for perturbed structures of individual molecules and how good the predictions are. What is the range of structural RMSD of the molecules considered in the table supplied on the reply to point 4 of reviewer 1? What do the subscripts "Global" and "R0.4" refer to? When writing "The reduced models show an improvement in prediction of the high-energy outliers": "improvement with respect to what"? How does a model on "extended structures" for a tetrapeptide perform on compact structures and vice versa?

3. The insight gained is still rather technical and the "key outcome" reads more like a conclusion for a journal specialized in computational or machine learning topics rather than providing deep(er) insight into a process or phenomenon. Maybe the authors can carry out (and validate) an MD simulation for a structural transition of the tetrapeptide which would illustrate the new possibilities.

Reviewer #2 (Remarks to the Author):

The authors responded satisfactorily to the criticisms and questions raised by the reviewers. I recommend publication.

Reviewer #3 (Remarks to the Author):

The authors have sufficiently addressed the concerns of the reviewers. The revisions have clarified several important details and resulted in an improved manuscript. I recommend that the manuscript is now suitable for publication.

Reviewer 1

Review of Tkatchenko et al. The authors have made several modifications and provided clarifications on points raised by the reviewers. A few points still remain to be addressed, nevertheless. Detailed comments:

1. What do the the authors refer to when writing "However, they all preserve the original energy distribution..", that the P(E) for the training, validation and test sets are similar/identical?

Yes, we mean that the energy distribution for training, validation, and test sets are identical.

2. It is still not clear whether features for individual structures for a database of molecules can be used for perturbed structures of individual molecules and how good the predictions are. What is the range of structural RMSD of the molecules considered in the table supplied on the reply to point 4 of reviewer 1? What do the subscripts "Global" and "R0.4" refer to? When writing "The reduced models show an improvement in prediction of the high-energy outliers": "improvement with respect to what"? How does a model on "extended structures" for a tetrapeptide perform on compact structures and vice versa?

We adapted abbreviations that we used in the paper, namely ML_{Global} - a GDML model trained with the original global descriptor and ML_{RX} - a GDML model trained with a reduced descriptor, where X denotes the size of the descriptor with respect to the original one (i.e. $ML_{R0.4}$ has descriptor with 40% of the features kept and 60% removed). We reiterated this information in the caption of Fig. 1 to make this clearer.

In the previous response, we showed that reduced models show an improvement in prediction of the high-energy outliers with respect to the global models for three studied systems. Now, for comparing how global/reduced models trained on "extended" structures perform on "compact" structures of the tetrapeptide, we have splitted the dataset based on the distance between the furthest atoms in each structure (ranges from ~ 8 to ~ 14 Å).

We selected a threshold of 12 Å which separates clusters of compact and extended structures. With this threshold, we splitted the dataset into dataset 1 ($\max(R_{ij}) < 12$ Å, $\sim 80\%$ of the initial dataset - 69k structures) and dataset 2 ($\max(R_{ij}) \geq 12$ Å, 20% - 16k structures). We used 1000 points for the training and 1000 for the validation of the models from the dataset 1 (training set - compact) and used all structures from the dataset 2 for testing (E/F RMSE extended).

To check how global/reduced models trained on "extended" structures perform on "compact" structures we repeated the same procedure with a threshold of 9.5 Å, resulting in the dataset 3 ($\max(R_{ij}) > 9.5$ Å, $\sim 80\%$ of the initial dataset) used for training (training set - extended) and dataset 4 ($\max(R_{ij}) \leq 9.5$ Å, 20%) used for testing (E/F RMSE compact). The results are presented in the table below, where the Energy RMSE is reported in kcal mol⁻¹ and the Force RMSE in kcal (mol Å)⁻¹.

Training set selected from	Model	E RMSE extended	F RMSE extended	E RMSE compact	F RMSE compact
compact	ML_{Global}	14.0	4.31	1.64	2.41
compact	$ML_{R0.4}$	7.55	3.55	1.47	2.24
extended	ML_{Global}	1.74	2.44	4.72	3.09
extended	$ML_{R0.4}$	1.53	2.29	3.05	2.67

The comparison of the Force/Energy RMSEs shows that the reduced models are more accurate than global models when dealing with "unseen" outlier structures. We can attribute such an improvement to the ability of reduced models to obtain a better description of the environments of the molecule. This means that reduced models can better identify similar structural moieties between "compact" and "extended" structures while keeping the relevant information for describing non-local interactions.

We also note that state-of-the-art local neural-network-based machine learning models produce worse results when testing in this scenario (these results go beyond our publication and we prefer to present this data in a forthcoming publication to avoid provoking unnecessary discussions at this moment).

The corresponding discussions have been added in the revised SI (pages S5-S6).

3. The insight gained is still rather technical and the "key outcome" reads more like a conclusion for a journal specialized in computational or machine learning topics rather than providing deep(er) insight into a process or phenomenon.

We respectfully disagree with this appreciation of the reviewer. Our work makes substantial breakthroughs in the broad domain of machine-learning force fields (MLFFs). These breakthroughs include (i) demonstrating the potential for linear scaling in global MLFFs for large systems, (ii) analyzing the non-local interatomic features that contribute to accurate predictions, and (iii) demonstrating the accuracy, efficiency, and stability of reduced models in long time-scale molecular dynamics simulations. Neither of these three achievements have been demonstrated to date in state of the art MLFFs. This, along with the very favorable assessments of Reviewers 2 and 3, make a strong case for the publication of our manuscript in *Nature Communications*.

Maybe the authors can carry out (and validate) an MD simulation for a structural transition of the tetrapeptide which would illustrate the new possibilities.

We have significantly improved the stability of the dynamics when employing ML models. For instance, in the case of the tetrapeptide, we were able to extend the simulation time from 0.3 ns to 50 ns by running 17 parallel simulations, each averaging around 3 ns. Our simulations captured both compact and extended structures, indicated by blue and red dots, respectively (see Fig. S4). To achieve this stability, we employed the reduced $ML_{R0.6}$ model trained on 5000 configurations with a shorter time step of 0.3 fs and a Langevin thermostat with a less aggressive $1 \cdot 10^{-4}$ friction coefficient.

Similarly, for the buckyball catcher, we used $ML_{R0.2}$ model trained on 1000 configurations with a timestep of 0.5 fs and the same thermostat to simulate the dynamics for 3 ns. In the case of AT-AT, we employed $ML_{R0.5}$ model trained on 1000 configurations with a shorter timestep of 0.1 fs to simulate the dynamics for 3 ns.

To further validate our reduced model, we performed steered MD simulations that forced the tetrapeptide molecule to extend by applying 10 pN restraints to the two terminal carbon atoms (Fig. 4, Fig. S3). Statistics were collected using 30 simulations with different initial velocities. We run simulations using the global and reduced models ($ML_{R0.6}$ trained with 5000 training points), as well as the simulations at the PBE and PBE+MBD level of theory for validation. We measured the structural compactness using the gyration radius and compared the dynamical properties of the models. As expected, the PBE simulations unfolded faster due to the absence of long-range interactions compared to the PBE+MBD simulations. Both the global and reduced models agreed well with the PBE+MBD results, indicating their accuracy and reliability. Also, this confidently shows that the reduced model preserves all the information needed to describe long-range interactions with *ab initio* accuracy.

We note that a reduced model trains 8-fold faster compared to the global model, and it is twice faster in evaluation speed.

The corresponding discussion has been added to the revised manuscript (lines 357-391).

Reviewer 2

The authors responded satisfactorily to the criticisms and questions raised by the reviewers. I recommend publication.

Reviewer 3

The authors have sufficiently addressed the concerns of the reviewers. The revisions have clarified several important details and resulted in an improved manuscript. I recommend that the manuscript is now suitable for publication.

REVIEWERS' COMMENTS

Reviewer #1 (Remarks to the Author):

Re-review of Kabylda et al.

The authors have clarified and extended their work. In particular the new results in Table S1 and Figures S3 and S4 are very convincing.

A few final remarks:

1. Table S1 needs to be mentioned and briefly discussed in the main manuscript.
2. Maybe add "on the 50 ns time scale" to "Our analysis reveals that the tetrapeptide populates the extended state with a probability of 13% (Fig. S5).".
3. The results in Table S1 appear to indicate that an overdetermined ("global") descriptor "confuses" the statistical model - is this a valid interpretation and if yes, can one state more generically that "overdetermined statistical models" are likely to underperform because of conflicting information?
4. The author's reply to my point 3, i.e. their key findings, read convincing and I suggest to include them explicitly into their discussion. Whether or not other reviewers comment favourably is of less concern.
5. Why not rephrase "However, they all preserve the original energy distribution.." to "The energy distributions for training, validation and test sets overlap." "Preserve" makes it sound as if there was something "active" in the background.

With the above points addressed the work is recommended for publication.

Reviewer 1 (Remarks to the Author):

Re-review of Kabylda et al.

The authors have clarified and extended their work. In particular the new results in Table S1 and Figures S3 and S4 are very convincing.

A few final remarks:

1. Table S1 needs to be mentioned and briefly discussed in the main manuscript.

We now discuss Table S1 in the main manuscript.

2. Maybe add "on the 50 ns time scale" to "Our analysis reveals that the tetrapeptide populates the extended state with a probability of 13% (Fig. S5).".

This suggestion is now addressed.

3. The results in Table S1 appear to indicate that an overdetermined ("global") descriptor "confuses" the statistical model - is this a valid interpretation and if yes, can one state more generically that "overdetermined statistical models" are likely to underperform because of conflicting information?

Noisy features can provide conflicting information. In that regard the referee is right.

4. The author's reply to my point 3, i.e. their key findings, read convincing and I suggest to include them explicitly into their discussion. Whether or not other reviewers comment favourably is of less concern.

We included our key findings in discussion explicitly as suggested by the reviewer.

5. Why not rephrase "However, they all preserve the original energy distribution.." to "The energy distributions for training, validation and test sets overlap." "Preserve" makes it sound as if there was something "active" in the background.

This suggestion is now addressed.

With the above points addressed the work is recommended for publication.

We sincerely thank the referee for his/her time in engaging with our work.